# Residual Stress Induced by Addition of Nanosized TiC in Titanium Matrix Composite

**DOI:** 10.3390/ma15072517

**Published:** 2022-03-29

**Authors:** Hanna Myalska-Głowacka, Bartosz Chmiela, Marcin Godzierz, Maria Sozańska

**Affiliations:** 1Faculty of Materials Engineering, Silesian University of Technology, Krasinskiego 8, 40-019 Katowice, Poland; bartosz.chmiela@polsl.pl (B.C.); maria.sozanska@polsl.pl (M.S.); 2Centre of Polymer and Carbon Materials Polish Academy of Sciences, Curie-Sklodowskiej 34, 42-819 Zabrze, Poland; mgodzierz@cmpw-pan.edu.pl

**Keywords:** composite, titanium, nanosized TiC, Ti–TiC, boron nitride, WDS, EBSD, residual stress analysis

## Abstract

A hot pressing process was employed to produce titanium-based composites. Nanosized TiC particles were incorporated in order to improve mechanical properties of the base material. The amount of nanosized additions in the composites was 0.5, 1.0, and 2.0 wt %, respectively. Moreover, a TiB phase was produced by in situ method during sintering process. The microstructure of the Ti–TiB–TiC composites was characterized by scanning electron microscopy (SEM), electron probe microanalysis (EPMA), electron backscatter diffraction (EBSD), and X-ray diffraction (XRD) techniques. Due to the hot pressing process the morphology of primary TiC particles was changed. Observed changes in the size and shape of the reinforcing phase suggest the transformation of primary carbides into secondary carbides. Moreover, an in situ formation of TiB phase was observed in the material. Additionally, residual stress measurements were performed and revealed a mostly compressive nature with the fine contribution of shear. With an increase in TiC content, linear stress decreased, which was also related with the presence of the TiB phase.

## 1. Introduction

Titanium matrix composites reinforced with ceramic particles are promising candidates in industrial applications, such as in the automotive, aerospace, biomedical, marine, and military industries. They are characterized by a high specific modulus and strength, and have excellent resistance to fatigue, wear, and high temperature creep compared to monolithic Ti alloys. Due to the excellent properties, titanium matrix composites can be applied on valves, connecting rods, and piston pins [1,2,3,4,5]. Interest in using nanostructured materials increased due to the favorable mechanical, optical, magnetic, and electrical properties of those structures. Nanostructured materials can be obtained in severe plastic deformation [6,7,8], nanocrystallization with metallic glasses as precursors [9,10], the implementation of nanoparticles [11,12], sol-gel method, hydrothermal synthesis, coprecipitation, and many others [13].

Ceramic materials are incorporated in order to improve strengthening effects in metal matrices using Orowan’s mechanism and thereby to enhance mechanical properties. Proper distribution of particles in a metal matrix is extremely important to achieve intended enhanced properties such as improved wear characteristics [14,15]. Nanoscale grain size in material plays an important role in microstructural changes during sintering at a faster rate due to the larger reaction area [16,17,18,19,20]. Composites reinforced with nanoceramic particles may be categorized into two groups that are mainly based on microstructural evaluation: (1) nanocomposites fabricated through the dispersion strengthening process of nanosized particles within micron-sized matrix grains or dispersed at the boundary of the grains within the matrix, and (2) nanometer scale with nanocomposites where both matrix grains and reinforcement are in the nanoscale [21,22].

As the most effective reinforcement, the following particles are considered: TiN, TiC, WC, Si_3_N_4_, SiC, TiB_2_, TiB, Al_2_O_3_, and carbon nanotubes (CNT) [15,16,23,24,25,26,27,28]. Extensive investigations were performed on (TiB + TiC)–Ti composites [2,29,30,31]. The composites were mainly in situ fabricated by the incorporation of B_4_C in Ti matrices via selective laser melting, spark plasma sintering, hot pressing, and hot extrusion [2,32,33,34]. Both TiB and TiC reinforcements exhibited good compatibility with a titanium matrix due to their comparable thermal expansion coefficients and densities [32,35]. Additives play a beneficial role on grain refining and hindering extensive grain growth, and are characterized by good bonding to the titanium matrix, consequently improving the mechanical properties of the composite [31]. Moreover, TiB can improve the creep resistance of titanium matrix composites, and reveals good chemical stability at very high temperatures with titanium without any reaction. Additionally, the phase ensures higher stiffness of the composite [36]. TiC enhances hardness and flow stress, and the uniform distribution of particles is considered to restrict dislocation movement [30]. The mechanical properties of the Ti matrix can be synergistically enhanced by TiB + TiC reinforcement [32,33].

Residual stresses created on the surface of machined components are one of the most crucial aspects influencing product performance. According to its attributes (compressive or tensile), these stresses can be either beneficial or detrimental to the performance of machined parts [37,38]. Machined components often exhibit tensile and/or compressive residual stresses, which can be induced by surface thermal gradients and/or by differential plastic deformation [39,40]. Tensile residual stresses in the surface and subsurface layers are usually detrimental to creep life, fatigue life, and stress corrosion cracking resistance, while compressive residual stresses are usually beneficial to the same properties [39,41,42,43,44].

Residual stress analysis is a powerful tool that allows for predicting the lifespan of many engineering materials, especially considering their mechanical properties. Many techniques can be applied and should be categorized on the basis of the degree of damage caused to the specimen. Thus, (1) destructive, (2) semidestructive, and (3) nondestructive residual stress measurement techniques exist [41,45,46]. In nondestructive methods, such as a neutron diffraction and X-ray diffraction XRD (sin2ψ method), crystal lattice strain is measured [46,47]. Then, corresponding residual stresses are estimated using elastic constants, assuming the linear elastic deformation of the crystal lattice. Nondestructive techniques can provide information on a series of material properties: tensile modulus, fracture toughness, microstructure and defects, etc [41].

In the present work, titanium composites reinforced with TiC were obtained by hot pressing. Microstructural and phase composition analysis revealed the presence of a TiB phase in the material. TiB was obtained by in situ formation due to the interaction among BN, TiC, and Ti. The aim of the study was to reveal the influence of microstructural changes of titanium composites reinforced with TiC caused by the presence of nanoparticles on internal stresses using the sin2ψ method, which has not been reported before.

## 2. Materials and Methods

Composites were obtained by hot pressing (HP) from titanium with titanium carbide powder mixtures. Titanium powder (KAMB Import-Export, Warsaw, Poland) with APS < 45 µm was mixed with nanosized TiC (Hefei EvNano Technology, Hefei, China). The amount of nanosized additions in the powder mixtures was 0.5, 1.0, and 2.0 wt %. Two stages of pressing in a Degussa press under vacuum were applied. In the first stage, samples were heated up to 400 °C with pressure of 1.5 MPa. In the second stage, pressure was 15 MPa, and temperature was 1200 °C. The hot-pressing conditions were applied for all powder mixtures. Moreover, as a reference sample, pure titanium powder also underwent the hot-pressing process. Samples were hot-pressed in graphite molds. In order to avoid interaction between powder and mold, a layer of h-BN was sprayed onto the mold and punch. The application of boron nitride increases sliding properties and allows for the easier removal of the sample from the mold.

Obtained composites were cut, and cross-sectional samples were investigated by scanning electron microscopy (SEM) and X-ray diffraction (XRD) techniques. Microstructural investigations were carried out with a S-3400N SEM (Hitachi, Tokyo, Japan) and Quanta FEI 250 FEG-SEM (ThermoFischer Scientific, Waltham, MA, USA). The applied acceleration voltage was 15 kV. Analyses of chemical composition were performed by electron probe microanalysis (EPMA) using energy-dispersive X-ray spectrometer (EDS) Thermo Noran (ThermoFischer Scientific, Waltham, MA, USA) and wavelength-dispersive X-ray spectrometer (WDS) Thermo MagnaRay (ThermoFischer Scientific, Waltham, MA, USA) operating at 15 keV of primary beam energy. Phase identification in microareas was performed by electron backscatter diffraction (EBSD) using INCA HKL Nordlys II detector (Oxford Instruments, Abingdon, Great Britain).

The microhardness of composites was determined on cross-sectional samples with the Vickers method and a Microhardness Tester FM-800 under indentation of 300 g (HV0.3).

The theoretical analysis of chemical interactions between elements was supported by thermodynamic data calculations obtained using Outotec’s HSC Chemistry 6.2 Software.

XRD analysis was performed using a D8 Advance diffractometer (Bruker, Karlsruhe, Germany) with Cu–Kα cathode λ = 1.54 Å) operating at 40 kV voltage and 40 mA current. Measurements were performed in the range of 2Θ = 20° ÷ 80° with 0.02° step size and 6.0°/min scan rate. Fitted phases were identified using DIFFRAC.EVA 5.1.0.5 (2019, Bruker AXS, Karlsruhe, Germany) software with the ICDD PDF#2 database, while the exact lattice parameters and crystallite size of fitted phase were calculated using Rietveld refinement in TOPAS 6.0.0.9 (2018, Bruker AXS, Karlsruhe, Germany) software on the basis of Williamson–Hall theory [48,49]. The pseudo-Voigt function was used in the description of diffraction line profiles at Rietveld refinement. Weighted-pattern factor R_wp_ and goodness-of-fit S parameters were used as numerical criteria of the quality of the fit of the calculated to the experimental diffraction data. The calculation of crystallite size using Rietveld refinement gives comparable results to crystallite size calculated using the Scherrer equation, but allows for simultaneously calculating lattice parameters, crystallite size, and lattice strain [50].

Residual stress analysis was performed using the isoinclination mode of the D8 Advance diffractometer (Bruker, Karlsruhe, Germany) with a Cu–Kα cathode (λ = 1.54 Å) operating at 40 kV voltage and 40 mA current with use of (213) peak of the α-Ti phase, according to the EN-15305 standard. RSA measurements were performed in three φ directions on the sinters’ cross-section to obtain a reliable stress mode. Φ = 0° was established to be parallel to longest dimension (perpendicular to pressing during sintering) [51]. Results were evaluated using DIFFRAC.LEPTOS 7.10.0.12 (2018, Bruker AXS, Karlsruhe, Germany) software, all peaks were fitted using standard fit, and applied stress mode was established to be triaxial.

## 3. Results

### 3.1. Powder Microstructure

Micrographs of the used initial powder are shown in Figure 1. The titanium powder consisted of elongated particles, and the surface of titanium granules was irregular, which might be a beneficial feature during the mixing process and the deposition of nanoparticles on a single granule. The nanosized TiC powder was characterized by very fine particles below 40 nm. The titanium carbide did not reveal a significant tendency to agglomerate.

XRD results of the initial powder are shown in Figure 2. Analysis confirmed that the titanium powder consisted only of a α-Ti phase (ICDD #00-005-0682), and the TiC powder only consisted of a TiC phase (ICDD 00-031-1400). The crystallite sizes of α-Ti and TiC powder (Table 1), calculated using Rietveld refinement, were 127.0 ± 5.0 and 77.0 ± 7.0 nm, respectively.

### 3.2. Composite Microstructure

EPMA results of the sintered Ti are shown in Figure 3. Some boron-containing particles were found in the microstructure, as WDS elemental mapping clearly revealed the presence of boron (Figure 3b). EDS mapping revealed a high relative concentration of titanium, what suggests that the particle belonged to the Ti–B system.

Phase identification performed by EBSD (Figure 4) confirmed that the observed particle was a titanium boride TiB (orthorhombic; space group 62; unit cell lengths: 0.612, 0.306, and 0.456 nm). The TiB phase was in the shape of needles and it was located in the entire volume of the sample.

Microstructural analysis of the Ti–0.5%TiC sinter revealed additional particles other than TiC. Moreover, TiC particles changed its morphology and size. TiC nanoparticles were no longer visible in the matrix, while needlelike TiC particles appeared. This may suggest a transformation from primary into secondary carbides. WDS elemental mapping showed the presence of carbon and boron (Figure 5). In all cases, boron-containing particles were placed very close to the titanium carbide particles and were also distributed in the entire volume of the sample. These observations may suggest a chemical reaction between BN and TiC particles during the sintering process.

EBSD analysis confirmed the presence of TiC (cubic; space group 225; unit cell lengths: 0.433, 0.433, 0.433 nm) and TiB (orthorhombic; space group 62; unit cell lengths: 0.612, 0.306, 0.456 nm; Figure 6).

The Ti–1.0%TiC sinter contained titanium carbide particles and particles with boron (Figure 7).

EBSD analysis revealed the presence of secondary titanium carbide Ti_5.73_C_3.72_ (hexagonal; space group 144; unit cell lengths: 0.306, 0.306, and 1.491 nm) and titanium boride TiB (orthorhombic; space group 62; unit cell lengths: 0.612, 0.306, 0.456 nm; Figure 8). Secondary carbide Ti_5.73_C_3.72_ was depleted with carbon in comparison to the primary carbide TiC.

The microstructure of the Ti–2.0%TiC sinter was similar to that of sinters with a lower concentration of TiC. Boron-containing particles were found close to the titanium carbides the (Figure 9).

Phase identification performed by EBSD showed the same secondary carbides as those in the Ti–1.0%TiC sinter: Ti_5.73_C_3.72_ (hexagonal; space group 144; unit cell lengths: 0.306, 0.306, and 1.491 nm) and titanium boride TiB (orthorhombic; space group 62; unit cell lengths: 0.612, 0.306, and 0.456 nm; Figure 10).

The presence of TiB particles in sintered titanium can be explained by the diffusion of BN into titanium, and the following chemical reaction between BN and Ti (1):2BN + 2Ti → 2TiB + N_2_(1)

Theoretical analysis of chemical interactions between Ti and BN can be supported by thermodynamic data calculations for different temperatures obtained using chemistry software HSC 6.2, and results are summarized in Table 2.

According to the thermodynamic data, TiB was formed due to the direct interaction between BN and Ti. The other reactions in Table 2 were thermodynamically much less likely due to more positive Gibbs energy values that were independent of the temperature of sintering. The thermodynamic calculations of Gibbs energy did not take pressure into account, but the equations of chemical reactions allow for some insights. The thermodynamically most likely reaction (2BN + 2Ti → 2TiB + N_2_) should be less efficient at higher pressure because one of the products (N_2_) is in gaseous form; therefore, increased pressure shifts the equilibrium of the reaction towards the reactants. Two other reactions characterized by negative Gibbs potential at higher temperature (2BN + Ti → TiB_2_ + N_2_ and 2BN + TiC → TiB_2_ + C + N_2_) may be less efficient at higher pressure due to the same reason.

The results of microhardness measurements are shown in Table 3. The lowest microhardness was detected for pure titanium sinter at the level of 301.5 HV_0.3_, while the highest value of 368.1 HV_0.3_ was calculated for the Ti–1%TiC composite. An increase in hardness was observed with an increase in TiC content in the material. However, after reaching some level of particles in the composite, microhardness started to slightly decrease (above 2 wt % of TiC).

### 3.3. Phase Composition of Composites

Obtained XRD results for all examined bulk materials are shown in Figure 11 and Figure 12, and Table 4. The sample produced from pure Ti powder was composed of hexagonal α-Ti (ICDD#00-005-0682), monoclinic TiO_2_ (ICDD#03-065-6429), and TiB (ICDD#01-073-2148). Moreover, the fine peak of pure graphite (ICDD#00-023-0064) was present in the sample. In the case of all composite Ti–TiC samples, an additional TiC phase was found (ICDD#00-031-1400), but no graphite was observed. A four-phase model of the material was used for Rietveld refinement, which gave a better goodness of fit (Figure 12).

The mean size of the α-Ti crystallite was 34 nm for all samples with small variation. This suggests a reaction during sintering due to the almost 4 times greater crystallite size in the used powder. During sintering, crystallite growth should occur unless reactive sintering is applied. Lattice parameters of that phase slightly changed, but were generally in good agreement with the model data, suggesting that no significant changes occurred in the matrix. However, the enlargement of the c parameter may suggest the presence of residual stress in the matrix. In the obtained composites, TiC particles revealed smaller crystallite size and lower lattice parameters than those in the initial TiC powder. In the Ti–TiC composites, the lattice parameters of TiC were 1–1.5% lower compared to the ICDD data. A finer crystallite size may suggest the reactive nature of the initial particles and the formation of secondary carbides. In the case of monoclinic TiO_2_, lattice parameters were in good agreement with the model data (ICDD data), while their crystallite size suggested a reactive nature without overgrowth of crystallites. More likely, TiO_2_ was formed prior to the sintering process (or in the first stage), and during this process, refinement occurred. For an orthorhombic TiB in the sinters, slight enlargement of *a* parameter was detected (up to 2.6%) with the simultaneous contraction of *c* parameter (1.3%). Such changes in lattice parameters indicated the reactive nature of TiB. In the pure Ti sinter, the TiB phase revealed almost perfect fitting to the ICDD data. Obtained results suggest an influence of TiC presence on TiB formation.

### 3.4. Residual Stress

Residual stress analysis performed on pure Ti sinter using diffraction line (213) indicated the compressive nature of the stress with fine contribution of shear. Detailed results are presented in Figure 13 and in Table 5. The compressive nature of residual stress is a result of the applied manufacturing process (hot pressing), and the presence of shear stress showed the displacement of powder during the pressing of the powder mixture. In all the obtained samples, residual stress was not linear, which indicated the inhomogeneity of the material. The highest linear stress in all samples was detected perpendicularly to pressing direction, while in parallel to the pressing direction, residual stress was the lowest. Inhomogeneity of residual stress was detected in the pure titanium sample, titanium with 0.5% of nanosized TiC, and titanium with 1% of nanosized TiC. This phenomenon suggests two factors: the high inhomogeneity of the material or texture presence. In the titanium composite with 2% of TiC, stress values were comparable in all examined directions.

The highest value of shear stress was detected in the Ti–0.5%TiC composite. For a higher amount of nanosized addition than 0.5%, shear stress decreased. This may have been the result of the reactive growth of TiB near TiC particles. Most likely, the formation of TiB is exothermic with a local increase in temperature, which allows for reducing shear stress in the titanium matrix.

## 4. Discussion

The formation of the TiB phase and secondary TiC precipitation could have been caused by the following effect. Due to its nature, boron atoms form complex low-energy interstitial defects (crowdions), and the transportation of these defects during a high-temperature forming process is activated. Then, the atoms might react with the titanium matrix, forming a TiB phase. Similar behavior can be observed for carbon atoms. A solid diffusion of carbon atoms occurred from TiC particles to the Ti matrix, and resulted in the formation of vacancies in the particle. Furthermore, carbon interacted with the matrix, and secondary TiC particles were formed [52]. Moreover, this process took place in the whole volume of the material. The diffusion processes can also explain the formation of carbon-depleted Ti_5.73_C_3.72_ phase particles.

Literature data suggest [53] that in situ formed TiB whiskers, and TiC particles formed clusters in the titanium matrix. The formation of TiB first starts due to higher energy of formation than that of TiC [54]. Furthermore, Horiuchi et al. [55] indicated that boron promotes a phase transformation from β to α in the (TiB + TiC)−Ti composites due to its capability to reduce the temperature of martensitic transformation.

Han et al. [53] performed atomic stress distribution analysis on selective laser-melted TiC–Ti samples. Two models demonstrated the accumulation of internal stress in the grain boundaries and at the tips of pores (cracks). Simulation results revealed that atomic stresses at the grain boundaries in the TiC–Ti model were lower than those in the Ti model. This phenomenon was caused by created vacancies (absence of carbon) that partially released the accumulated stress in the TiC particles.

In the conducted research, the following phenomena were observed. An increase in TiC addition decreased linear stress and increased shear stress. Additionally, greater content of TiC induced a greater diffusion process and consequently increased the TiB phase content, and the stress field became more homogenous independently of the sample orientation. The presence of TiB additionally influenced residual stresses in the material. The formation of TiB probably reduced the compressive stress and increased the homogeneity of the linear stress due to two factors: local temperature increase (highly exothermic formation reaction) [2,56] and/or excellent sliding properties of TiB [57], which may act as a solid lubricant during titanium powder movement during the sintering process. Moreover, TiC caused the refinement of TiB crystallites, which may have confirmed the assumptions that TiC particles introduce compressive stress, while TiB reduces this type of stress.

## 5. Conclusions

In this study, TiC–Ti composites were fabricated via hot pressing in a Degussa press using mixtures of pure Ti with 0.5%, 1.0%, and 2.0% of nanosized TiC. The following general conclusions can be drawn:The boron nitride applied as a protective layer prevented interaction between the graphite mold and titanium diffused into the composites, and caused an in situ formation of TiB phase.Due to the diffusion processes, the nanosized TiC particles incorporated into the titanium matrix reacted and formed secondary carbides.The hot-pressing processes resulted in the refinement of the Ti crystallite size by almost four times, regardless of whether TiC was introduced into the composite.With an increase in TiC amount in the composite, the crystallite size of secondary carbides slightly increased from 18 to 23 nm.The incorporated TiC caused the refinement of TiB crystallite size.Residual stress results mostly had a compressive nature with a fine contribution of shear, which is strongly related with the manufacturing process. With an increase in TiC content, linear stress decreased, which was also related to the presence of a TiB phase.

## Figures and Tables

**Figure 1 materials-15-02517-f001:**
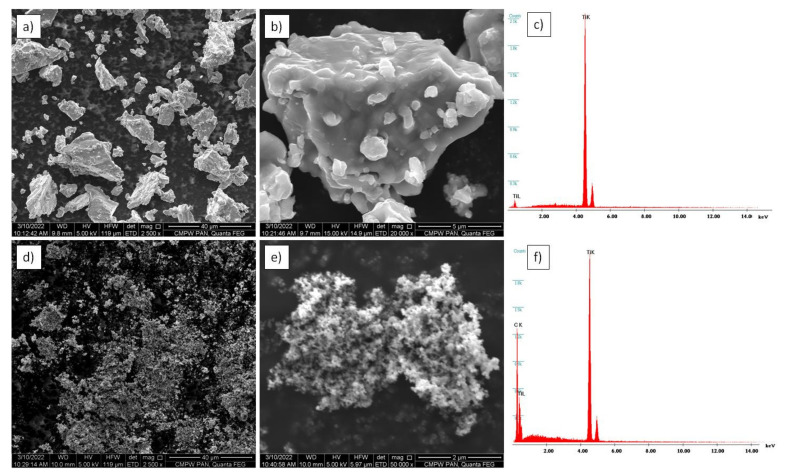
Morphology of initial powder: (**a**,**b**) titanium powder microstructure; (**c**) titanium powder EDX spectra; (**d**,**e**) nanosized TiC powder microstructure; (**f**) nanosized TiC EDX spectra, SEM.

**Figure 2 materials-15-02517-f002:**
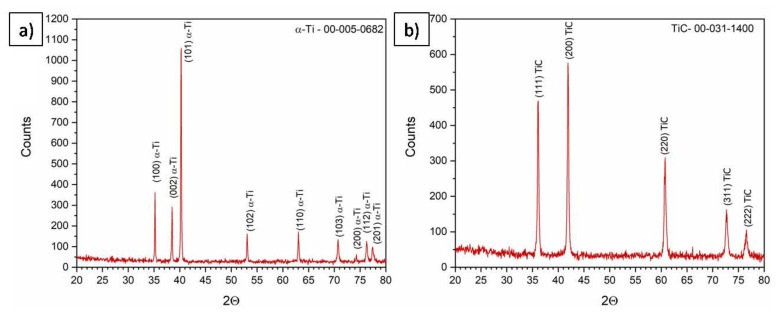
Phase composition of initial powder: (**a**) Ti powder; (**b**) TiC powder, XRD.

**Figure 3 materials-15-02517-f003:**
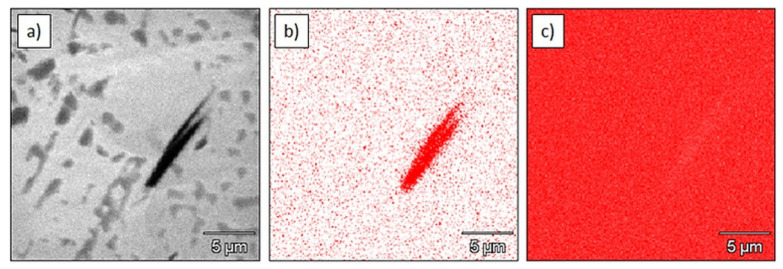
Microstructure and elemental maps of Ti sinter: (**a**) microstructure; (**b**) elemental map of boron (WDS); (**c**) elemental map of titanium (EDS).

**Figure 4 materials-15-02517-f004:**
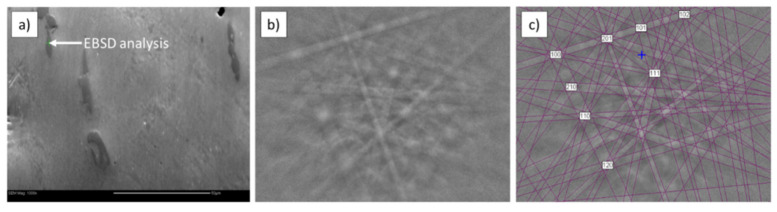
EBSD analysis of Ti sinter: (**a**) microstructure; (**b**) Kikuchi pattern of TiB phase; (**c**) indexed Kikuchi pattern of TiB phase.

**Figure 5 materials-15-02517-f005:**
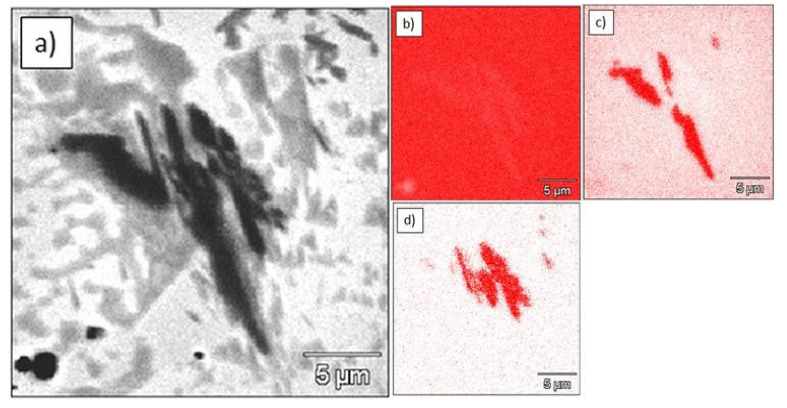
Microstructure and elemental maps of Ti-0.5%TiC sinter: (**a**) microstructure; (**b**) elemental map of titanium (EDS); (**c**) elemental map of carbon (WDS); (**d**) elemental map of boron (WDS).

**Figure 6 materials-15-02517-f006:**
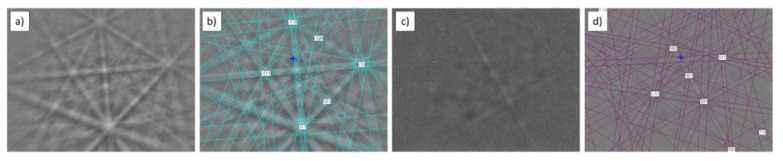
EBSD analysis of Ti–0.5%TiC sinter: (**a**) Kikuchi pattern of TiC phase; (**b**) indexed Kikuchi pattern of TiC phase; (**c**) Kikuchi pattern of TiB phase; (**d**) indexed Kikuchi pattern of TiB phase.

**Figure 7 materials-15-02517-f007:**
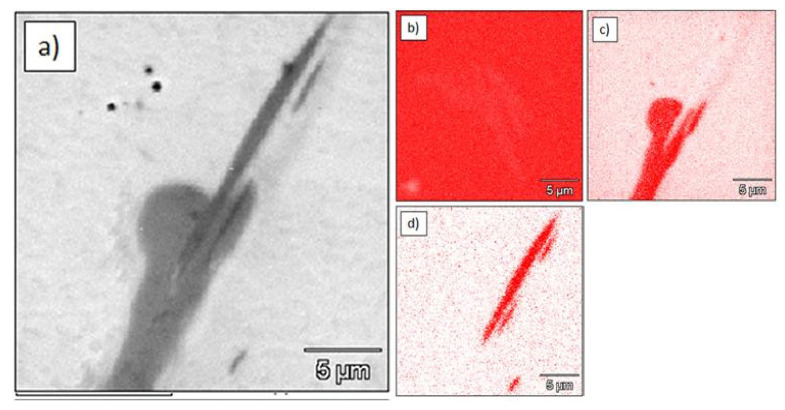
Microstructure and elemental maps of Ti-1%TiC sinter: (**a**) microstructure; (**b**) elemental map of titanium (EDS); (**c**) elemental map of carbon (WDS); (**d**) elemental map of boron (WDS).

**Figure 8 materials-15-02517-f008:**

EBSD analysis of Ti-1.0%TiC sinter: (**a**) Kikuchi pattern of Ti_5.73_C_3.72_ phase; (**b**) indexed Kikuchi pattern of TiC phase; (**c**) Kikuchi pattern of TiB phase; (**d**) indexed Kikuchi pattern of TiB phase.

**Figure 9 materials-15-02517-f009:**
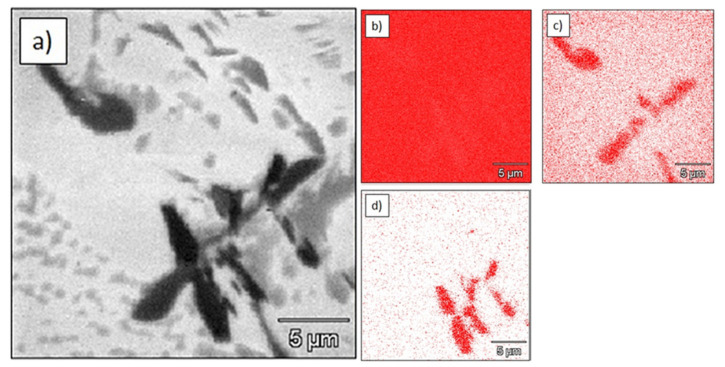
Microstructure and elemental maps of Ti-2.0%TiC sinter: (**a**) microstructure; (**b**) elemental map of titanium (EDS); (**c**) elemental map of carbon (WDS); (**d**) elemental map of boron (WDS).

**Figure 10 materials-15-02517-f010:**

EBSD analysis of Ti-2.0%TiC sinter: (**a**) Kikuchi pattern of Ti_5.73_C_3.72_ phase; (**b**) indexed Kikuchi pattern of Ti_5.73_C_3.72_ phase; (**c**) Kikuchi pattern of TiB phase; (**d**) indexed Kikuchi pattern of TiB phase.

**Figure 11 materials-15-02517-f011:**
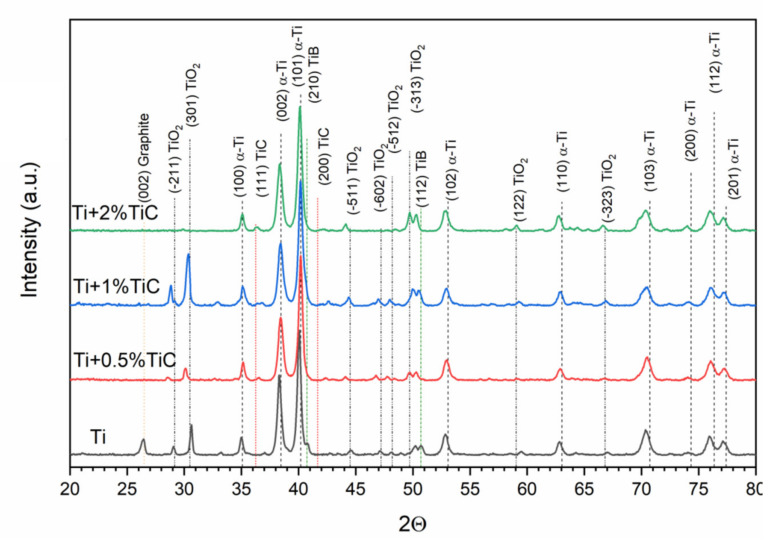
Phase composition of obtained titanium composites, XRD.

**Figure 12 materials-15-02517-f012:**
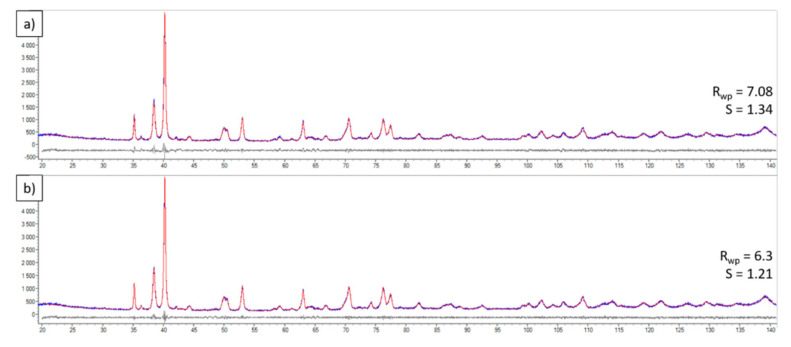
XRD patterns of Ti–2%TiC with differential curve calculated for (**a**) three-phase material (α-Ti, TiO_2_, TiC) and (**b**) four-phase material (α-Ti, TiO_2_, TiC and TiB).

**Figure 13 materials-15-02517-f013:**
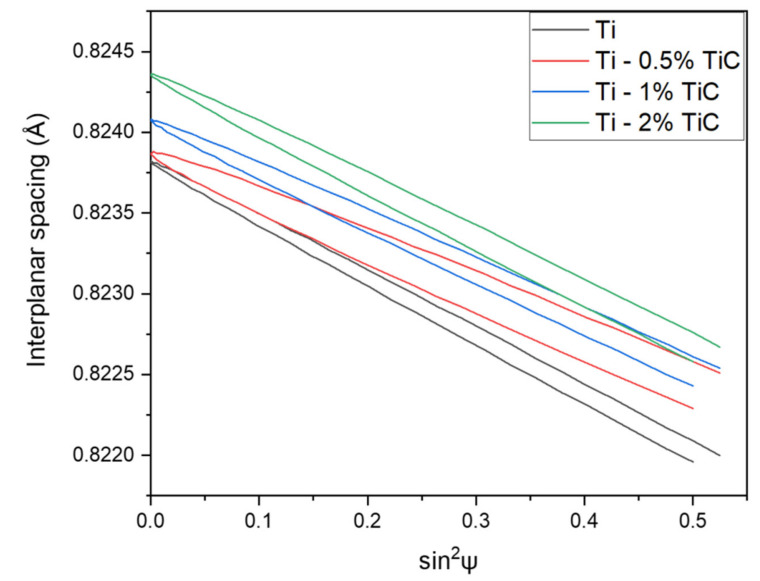
Examples of residual stress diagrams obtained for Ti and Ti–TiC sinters.

**Table 1 materials-15-02517-t001:** Lattice parameters and crystalline size of initial powder calculated using Rietveld refinement.

Sample	Phase	Space Group	Calculated Lattice Parameters, Å	Lattice Parameters, ICDD, Å	Crystallite Size, nm
Ti powder	α-Ti (00-005-0682)	P6_3_/mmc	a = 2.951 c = 4.686	a = 2.95 c = 4.686	127.0 ± 5.0
TiC powder	TiC (00-031-1400)	Fm-3m	a = 4.313	a = 4.33	77.0 ± 7.0

**Table 2 materials-15-02517-t002:** Gibbs free energy for selected reactions in Ti–C–B–N system at temperatures of 400 and 1200 °C.

Reaction	Gibbs Potential ΔG (kJ)	
	400 °C	1200 °C
Ti + C → TiC	−176.7	−166.8
Ti + B → TiB	−158.5	−153.2
Ti + TiB_2_ → 2TiB	−47.5	−53.1
2BN + 2Ti → 2TiB + N_2_	−269.5	−253.3
2BN+ Ti → TiB_2_ + N_2_	65.7	−64.5
2BN + TiC → TiB_2_ + C + N_2_	113.3	−11.4
2BN + 2TiC → 2TiB + 2C +N_2_	290.0	155.4

**Table 3 materials-15-02517-t003:** Microhardness results of obtained composites.

Sample	Ti	Ti–0.5%TiC	Ti–1.0%TiC	Ti–2.0%TiC
Microhardness, HV0.3	301.5 ± 41.8	337.8 ± 54.0	368.1 ± 76.7	361.9 ± 41.8

**Table 4 materials-15-02517-t004:** Lattice parameters and crystalline size of identified phases calculated using Rietveld refinement.

Sample	Phase	Space Group	Lattice Parameters, Calculated, Å	Lattice Parameters, ICDD, Å	Crystallite Size, nm
Ti	α-Ti (00-005-0682)	P6_3_/mmc	a = 2.96 c = 4.71	a = 2.95 c = 4.686	30.0 ± 9.0
	Graphite (00-023-0064)	P6_3_/mmc	a = 2.45 c = 6.76	a = 2.465 c = 6.721	27.0 ± 3.0
	TiO_2_ (03-065-6429)	P2_1_/m	a = 12.20 b = 3.73 c = 6.53 β = 107.19°	a = 12.1787 b = 3.7412 c = 6.5249 β = 107.05	32.0 ± 9.0
	TiB (01-073-2148)	Pnma	a = 6.13 b = 3.06 c = 4.55	a = 6.12 b = 3.06 c = 4.56	46.0 ± 3.0
Ti–0.5%TiC	α-Ti (00-005-0682)	P6_3_/mmc	a = 2.95 c = 4.70	a = 2.95 c = 4.686	36.0 ± 5.0
	TiC (00-031-1400)	Fm-3m	a = 4.264	a = 4.33	18.0 ± 2.0
	TiO_2_ (03-065-6429)	P2_1_/m	a = 12.30 b = 3.71 c = 6.53 β = 107.29°	a = 12.1787 b = 3.7412 c = 6.5249 β = 107.05°	28.0 ± 15.0
	TiB (01-073-2148)	Pnma	a = 6.29 b = 3.04 c = 4.49	a = 6.12 b = 3.06 c = 4.56	22.0 ± 8.0
Ti–1.0%TiC	α-Ti (00-005-0682)	P6_3_/mmc	a = 2.96 c = 4.72	a = 2.95 c = 4.686	31.0 ± 2.0
	TiC (00-031-1400)	Fm-3m	a = 4.284	a = 4.33	13.0 ± 2.0
	TiO_2_ (03-065-6429)	P2_1_/m	a = 12.28 b = 3.70 c = 6.51 β = 107.33°	a = 12.1787 b = 3.7412 c = 6.5249 β = 107.05°	31.0 ± 9.0
	TiB (01-073-2148)	Pnma	a = 6.27 b = 3.06 c = 4.50	a = 6.12 b = 3.06 c = 4.56	19.0 ± 18.0
Ti–2.0%TiC	α-Ti (00-005-0682)	P6_3_/mmc	a = 2.96 c = 4.72	a = 2.95 c = 4.686	37.0 ± 3.0
	TiC (00-031-1400)	Fm-3m	a = 4.276	a = 4.33	23.0 ± 4.0
	TiO_2_ (03-065-6429)	P2_1_/m	a = 12.31 b = 3.70 c = 6.54 β = 107.07°	a = 12.1787 b = 3.7412 c = 6.5249 β = 107.05°	25.0 ± 10.0
	TiB (01-073-2148)	Pnma	a = 6.28 b = 3.04 c = 4.51	a = 6.12 b = 3.06 c = 4.56	30.0 ± 3.0

**Table 5 materials-15-02517-t005:** Residual stress values depending of sample orientation (φ angle).

Sample	Linear Stress, MPa			Mean Shear Stress, MPa
	0°	45°	90°	
Ti	−215.5 ± 30.4	−150.6 ± 25.7	−67.4 ± 30.4	5.4 ± 2.6
Ti–0.5%TiC	−108.9 ± 37.7	−71.4 ± 37.7	−35.9 ± 31.8	12.1 ± 6.2
Ti–1.0%TiC	−186.5 ± 27.8	−35.6 ± 23.5	−4.6 ± 27.8	7.8 ± 2.1
Ti–2.0%TiC	−92.7 ± 32.9	−80.0 ± 32.9	−40.3 ± 27.9	8.0 ± 6.5

## Data Availability

Not applicable.

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
