# Peer review of "Residual Stress Induced by Addition of Nanosized TiC in Titanium Matrix Composite"

_materials, 2022, doi:10.3390/ma15072517_

Round 1
Reviewer 1 Report
Review report on the topic ‘Residual Stress Induced by Addition of Nanosized TiC in Ti-tanium Matrix Composite’. Comments are listed below:
- A common type of information was presented in the abstract section. The abstract represents the summary of the work. Revise it carefully and add a key conclusion of the work.
- The novelty of the work should be discussed first in respect of the application.
- Try to strengthen the introduction section as many works have already been published on a similar topic. Try to make a bridge between current and previously published work and specify the gap area and objective of the work. Discuss some novel approaches used for residual stress measurement and also provide more discussion related to metal matrix composite: https://doi.org/10.1007/s40430-017-0757-1; https://doi.org/10.3390/ma14216591; https://doi.org/10.1016/j.matlet.2020.128347;
- The role of Ti addition is not clear.
- The literature section is very poor. Refer to some recently published work on MMC their characteristic and residual stresses evolution.
- It is difficult to reach any conclusion from Fig. 1b. Provide good quality SE image along with EDS spectra.
- The XRD results are presented without any discussion. Try to map the phase percentage using XRD peaks.
- Improve the SEM image quality (Fig. 4, 5, 6 and 8).
- Each Fig can be summarized in one combined Figure for better presentation. The aim is to represent the addition and increase in weight percentage of Ti. So please combine all the images.
- How is the Gibbs free energy measurement performed? Add more references in support of statements.
- Try to relate the residual stress variation with Ti addition.
- The discussion section is very poor, and it looks like a technical report. Add more discussion in support of results along with proper references.
Overall, the work is quite good and interesting and can be accepted after minor corrections.
Author Response
Dear Reviewer,
We wish to thank you for all your constructive comments in this round of review. Your comments provided valuable insights to refine its contents and analysis. In this document, we try to address the issues raised as best as possible.
Sincerely,
Hanna Myalska-GÅ‚owacka

Reviewer 2 Report
In this study TiC-Ti composites were fabricated via hot pressing in Degussa press using mixtures of pure Ti with 0.5%, 1% and 2% of nano-sized TiC. A hot pressing process was employed to produce titanium-based composites. Nano-sized TiC particles were incorporated in order to improve mechanical properties of the base material. The boron nitride applied as a protective layer preventing interaction between the graphite mold and titanium diffused into the composites and caused an in situ formation of TiB phase.
The manuscript is well-written and is suitable for publications in materials after major revision.
Objectives of the work is not clear list as bullet points at the end of introduction section.
Follow uniform decimal format throughout the text including abstract (single decimal or double).
In introduction section, recent articles in MATERIALS should be cited.
The SEM of nano-sized TiC is not clear, it is better to improve quality (if possible).
You may strengthen the literature by, effect of cutting parameters on surface residual stresses in dry turning of aisi 1035 alloy, prediction of residual stresses in turning of pure iron using artificial intelligence-based methods, fine-tuned artificial intelligence model using pigeon optimizer for prediction of residual stresses during turning of inconel 718, a comprehensive review on residual stresses in turning.
Why do in situ formed TiB whiskers and TiC particles 304form clusters in the titanium matrix?
You used ICDD#00-005-0682 and ICDD#00-031-1400 in the discussion, could you add the reference.
Any observations on the chemical reaction between BN and TiC in sintering process.
Why do the residual stress results have mostly compressive nature with fine contribution of shear?
The following paragraph should be rewritten: In the conducted research following phenomenon were observed. The linear stress decreases with an increase of TiC addition, while the shear stress increases. Additionally, the greater content of TiC, the greater TiB content in the composites. The stress field become more homogenous independently on the sample orientation. Moreover, a sliding properties typical for TiB may have additional influence on residual stress in the material. Probably it reduces the compressive stress and increases the homogeneity of the linear stress. The TiC causes refinement of the TiB crystallites, which may confirm the assumptions that the TiC particles introduce compressive stress, while TiB reduces this type of stress.
Author Response

(The authors gave the same response as above.)

Reviewer 3 Report
The authors present a work entitled “Residual Stress Induced by Addition of Nanosized TiC in Titanium Matrix Composite”. The authors obtained titanium composites reinforced with TiC by hot pressing. They revealed an influence of microstructural changes of titanium composites reinforced with TiC caused by the presence of nanoparticles on internal stresses. This study is interesting, but there are still some improvements to be made before it can be accepted for publication.
- Introduction, the authors should provide more descriptions about nanostructured materials, such as common preparation techniques of nano-structures (severe plastic deformation, nanocrystallization with metallic glasses as precursors, etc.). Please add appropriate references here. Below are a few references to help with this process.
[1] “Densification mechanism of Zr-based bulk metallic glass prepared by two-step spark plasma sintering”, Journal of alloys and compounds, 2020, 850:156724. 10.1016/j.jallcom.2020.156724
[2] “Enhancing strength-ductility synergy in an ex situ Zr-based metallic glass composite via nanocrystal formation within high-entropy alloy particles”, Material and design, 2021, 210:110108. 10.1016/j.matdes.2021.110108
Introduction, The aim of the study was to reveal an influence of the presence of nanoparticles on internal stresses. However, the authors did not make clear why we should pay attention to the residual stress, and the influence of residual stress on the microstructure and mechanical properties of Ti matrix composites. Relevant research advances in these areas should be added here.
- Line 281, The authors attribute the inhomogeneity of residual stress to the high inhomogeneity of the material or texture presence. The more nano-sized TiC is added, the more uneven the material should be. But in the titanium composite with 2% of TiC, why the stress values are comparable in all examined directions, that is more homogeneous residual stress? Please add a more comprehensive discussion and provide some research and experimental evidence to support the author's views.
- The boron nitride diffused into the composites and caused an in situ formation of TiB phase. Is it possible to produce a concentration gradient of element B in the nano-sized TiC reinforced Ti matrix composite sample? And furtherly, whether it may lead to changes in TiB content and residual stress along the diffusion direction of the B atoms? Whether the authors observed the above phenomena during their study? Please add some discussions about this.
Author Response

(The authors gave the same response as above.)

Round 2
Reviewer 2 Report
Accept in present form.
Reviewer 3 Report
Nice revision. Publish it.